# Water-promoted C-S bond formation reactions

Peizhong Xie[1], Jinyu Wang[1], Yanan Liu[1], Jing Fan[1], Xiangyang Wo[1], Weishan Fu[1], Zuolian Sun[1] & Teck-Peng Loh[1,2]

Allylic sulfones, owning to their widespread distributions in biologically active molecules, received increasing attention in the past few years. However, the synthetic method under mild conditions is still a challenging task. In this paper, we report a sulfinic acids ligation with allylic alcohols via metal-free dehydrative cross-coupling. Both aliphatic and aromatic sulfinic acids react with various allylic alcohols to deliver the desired allylic sulfones in high yields with excellent selectivity. This carbon–sulfur bond formation reaction is highly efficient and practical since it works under metal-free, neutral, aqueous media and at room temperature in which the products even can be obtained by simple filtration without the need for organic extraction or column chromatography. Water is found to be essential for the success of this carbon–sulfur bond formation reaction. DFT calculations imply that water acts as promoter in this transformation via intermolecular hydrogen bonds.

[1] School of Chemistry and Molecular Engineering, Institute of Advanced Synthesis, Jiangsu National Synergetic Innovation Center for Advanced Materials, Nanjing Tech University, Nanjing 211816, China. [2] Division of Chemistry and Biological Chemistry, School of Physical and Mathematical Sciences, Nanyang Technological University, Singapore 637371, Singapore. Correspondence and requests for materials should be addressed to P.X. (email: peizhongxie@njtech.edu.cn) or to T.-P.L. (email: teckpeng@ntu.edu.sg)

The development of atom economical organic reactions that can be carried out under metal-free, neutral, in aqueous media, and at room temperature in which the products can be obtained after the reaction by simple filtration without the need for organic extraction or column chromatography is one of the ultimate goals in organic synthesis. In our continuing effort to develop green processes, we focus on the development of an efficient and practical method for the synthesis of sulfones since they play a prominent role in organic synthesis and also exist widely in many bioactive natural products as well as some of the best-selling pharmaceuticals[1] (such as Eletriptan ([2H]-SB-3CT) for the treatment of migraine headache and MMP-2, MMP-9 inhibitor[2] for the treatment of prostate cancer). In this context, the development of efficient methods for the synthesis of allylic sulfones has attracted tremendous attention in the synthetic community[3]. Moreover, the wide distribution of this moiety in biologically and pharmaceutically active molecules[4–8], such as anticancer agents[4], cysteine protease inhibitors[5], antibacterial agents[6], weedicide[7], etc.[8] (Fig. 1), have also attracted considerable attention. Therefore, tremendous effort has been directed towards the development of efficient methods to access this class of compounds. Reported methods included the use of transition-metal catalyzed (oxidative)-cross-coupling[9] reactions, Tsuji–Trost reaction[10], and hydrosulfination[11] by using the highly reactive allylic substrates and sulfinyl nucleophiles. Coupled with environmental and economy concerns, the development of energy-efficient and greener synthetic methods from nontoxic, inexpensive, readily available and environmentally benign feedstock is extremely attractive and essential.

The direct use of synthetically reliable allylic alcohols that give water as by-product is particularly attractive in terms of step economy and positive environmental impact[12–15]. However, compared with the activated analogs such as sulfonates, ester, and ethers, the remarkable activation barrier of C–OH scission (C–O bond dissociation energy 85–91 kcal mol$^{-1}$[16]) and polarizability of O–H bond posed great challenge in their utility in allylic alkylation reactions. To address these problems, appropriate transition-metal, high reaction temperature and, quantitative addition of extra additive were generally required. By introducing stoichiometric amount of B(OH)$_3$/Et$_3$B or TMSCl, Tian[17, 18], Sreedhar[19], and Chandrasekhar[20] succeeded in developing Pd or Fe catalyzed promising allylic sulfonylation with allylic alcohols or amines (Fig. 2a). Pioneering investigation on the metal-free strategies for aryl sulfonylation of specific allylic alcohols at high temperature via nucleophilic addition[21] or carbocation process[22] was reported gradually. Unfortunately, the substrate scope was rather limited and only works with aromatic sulfones. The remarkable breakthrough for sulfonylation of broad-spectrum

allylic alcohols at ambient temperature was realized by Reddy et al. by utilizing the ArSO$_2$CN as sulfonyl nuclephiles (Fig. 2b)[23]. Nevertheless, this system still suffers from limitations inherent to quantitative amines, toxic and expensive reagents/solvent. With our continuing interest in green chemistry[24–26] and developing new bioconjugation method for cysteine[27], we embarked on developing an effective metal-free ligation reaction that selectively converts sulfinic acid moieties into allylic sulfones in aqueous media at room temperature under physiological pH. Theoretically, this process offers easy access to allylic sulfones in a green manner and also shed light on protein sulfinylation detection, including nitrile hydratase, matrilysin, and Parkinson's disease protein DJ-1[28, 29].

## Results

**Reaction condition optimization for synthesis of 3.** With the expectation to obtain trisubstituted allylic sulfones 3, which have been recently found to be highly potent against cancer and abnormal cell proliferation disease[4], we selected Morita–Baylis–Hillman (MBH) alcohol **1a** and benzenesulfinic acid **2a** as model substrates. To our delight, the desired product **3a** can be detected in 14% yield when the reaction was conducted in toluene at 80 °C without catalyst (Supplementary Table 1, Entry 1). With the attempt to improving the yield, the effect of solvent was screened. THF, 1,4-dioxane, and alcohols disfavored this reaction, which delivered only a trace amounts of **3a** (Supplementary Table 1). In our experience, much more positive result was obtained when the reaction proceeded without solvent (Supplementary Table 1, entry 9). We hypothesize that the water might affect this reaction. When the reaction was conducted in deionized water, the solubility is poor and 25% of **3a** was detected (Supplementary Figure 1a). Coupled with environmental concerns, co-solvent H$_2$O/EtOH (v/v = 1/1) was chosen to increase the solubility. To our great delight, the desired product **3a** precipitated from the solvent after the reaction and can be isolated by simple filtration in 75% yield (91% yield for column chromatography) (Supplementary Figure 1c). Satisfactory yield was achieved when co-solvent H$_2$O/CH$_3$CN (v/v = 1/1) was employed. It should be noted that the reaction also proceeded well at ambient temperature. Comparable results can be obtained even if this reaction is conducted in plastic tube (Supplementary Figure 1d) or PBS (pH 4.3)/EtOH (v/v = 1/1) solvent. H$_2$O/DMSO (v/v = 1/1) delivered the comparable result (Supplementary Table 2). Taking all the aforementioned results into account, it is reasonable to assume that the water itself is crucial for this transformation and the effect of co-solvent such as acetonitrile, ethanol, and dimethyl sulfoxide was just to enhance the solubility of the starting materials. Furthermore, no desired product was

**Fig. 1** allylic sulfones moiety in biologically active molecules. **a** Anticancer agents and anti-abnormal cell proliferation. **b** TSH receptor antagonists. **c** Anti-inflammatory and antidegenerative agents. **d** Cysteine protease inhibitors. **e** Antibacterial agents

**a** Metal-catalyzed direct substitution of allylic amines and alcohols. ref. (17-20)

X = NH, O

**b** Amine-promoted reaction between allylic alcohols and ArSO₂CN. ref. (23)

R' = CO₂R, COR, CN, H

**c** Water-promoted dehydrative Cross-Coupling (This work)

R¹ = Alkyl, Aryl, H; R² = CO₂R, COR, CN, Alkyl, H; R³, R⁴, R⁵ = Aryl, Alkyl

- ○ Environmental benign manner
- ○ Metal-free
- ○ Simple filtration without chromatography can isolate desired products
- ○ Aqueous media (pH7)
- ○ Room temperature

**Fig. 2** Strategies for direct sulfonylation of allylic alcohols. **a** Transition-metal (Pd and Fe)-catalyzed direct substitution of allylic amines and alcohols. Acidic additives (stoichiometric amount of B(OH)₃/Et₃B or TMSCl). **b** Amine (DIPEA: *N*-ethyl-*N*-isopropylpropan-2-amine) promoted reaction between allylic alcohols and ArSO₂CN. **c** In this report, water-promoted dehydrative cross-coupling reaction. The reaction conducted in an environmental benign manner under physiological conditions (pH 7, aqueous media, metal-free, and room temperature)

observed when the benzenesulfinic acid was replaced by sodium benzenesulfinate, which means the reaction worked not as a simple nucleophilic addition. The structure of **3a** was also determined by X-ray analysis (Supplementary Figure 78).

**Substrate scope**. With an optimized set of reaction conditions, the substrate scopes of the dehydrative cross-coupling reaction was investigated. The results are shown in Table 1. A variety of sulfinic acids reacted effectively to afford the desired products in high yield with excellent selectivity. Both electron-rich and eletron-deficient phenyl sulfinic acids could react with **1a**, producing the corresponding allylic sulfones in excellent yields (**3a–3c**). It should be noted that the alkanesulfinic acids, which had posed great challenge to be incorporated into allylic sulfones due to their lower reactivities[22], were suitable substrates in our reaction system (**3d, 3e**). The electronic property and position of the aryl substituents on MBH alcohols have limited effect on the overall performance of this transformation (**3f–3o**). Even the allylic alcohols bearing high active group such as cyano (**3k**) and formyl group (**3l**) could give the desired products in moderated yield. MBH alcohols bearing naphthalene-1-yl or heteraryl substituents thiophen-2-yl was also tolerated well to generate the

synthetically useful allylic sulfones (**3p, 3q**) with excellent performance associated with yield and selectivity. Alkyl-substituted MBH alcohol was suitable substrate for this reaction, albeit give the **3s** in moderate yield. Moreover, γ-blocked MBH alcohol **2t** can be succeeded in giving corresponding **3t** in 90% yield with good selectivity. The water-promoted dehydrative cross-coupling reaction also can be performed well in gram scale, and **3f** can be conveniently isolated by simple filtration.

Simple alcohols besides the MBH version were then investigated under the identical reaction conditions (Table 2). Both aliphatic and aromatic sulfinic acids reacted well with (*E*)-1,3-diarylprop-2-en-1-ol to produce the corresponding branched *E*-allylic sulfones in high yields (**3u–3y, 3af–3ah**). Even in gram scales synthesis, no chromatographic purification was necessary (**3u**). In addition, aryl substituents on allylic alcohols have little effect on the yields (**3z–3ac**). That means our developed process can be performed well with broad substrate-spectrum. α-blocked allylic alcohol can also be incorporated into this reaction, albeit affording **3ad** in lower yield. As expected, 1,1-diphenylprop-2-en-1-ol was suitable coupling partners in the title reaction and led to the **3ae** in good yield. Moreover, Stiripentol (Diacomit) (ME-2080), an anticonvulsant drug used in the treatment of epilepsy,

## Table 1 Substrate scope of Morita–Baylis–Hillman alcohols

**3a**: Z/E > 95/5, 91% yield

**3b**; Z/E > 95/5, 96% yield

**3c**; Z/E > 95/5, 97% yield

**3d**; Z/E > 95/5, 62% yield

**3e**; Z/E > 95/5, 82% yield

**3f**; Z/E > 95/5, 75% yield

**3g**; Z/E > 95/5, 75% yield

**3h**; Z/E > 95/5, 74% yield

**3i**; Z/E > 95/5, 72% yield

**3j**; Z/E > 95/5, 84% yield

**3k**: Z/E > 95/5, 67% yield

**3l**: Z/E > 95/5, 59% yield

**3m**; Z/E > 95/5, 82% yield

**3n**; Z/E > 95/5, 91% yield

**3o**; Z/E > 95/5, 72% yield

**3p**; Z/E > 95/5, 93% yield

**3q**; Z/E > 95/5, 51% yield

**3r**; Z/E > 95/5, 32% yield

**3s**; Z/E > 95/5, 44% yield

**3t**; Z/E > 95/5, 90% yield

**3a:** CCDC:1573430

**Gram scale (3f: 3.0g)**, Z/E > 95/5, **73% yield (by filtration)**, *without chromatography*

Experimental conditions: **1** (0.3 mmol) and **2** (0.45 mmol) dissolved in $H_2O$/EtOH (v/v = 1/1) (2.0 mL) and then stirred at 30 °C, isolated yields, *Z/E* was determined by [1]H NMR, It is a stereoselective rather than stereospecific reaction

**Table 2 Substrate scope of the dehydrative cross-coupling with allylic alcohols**

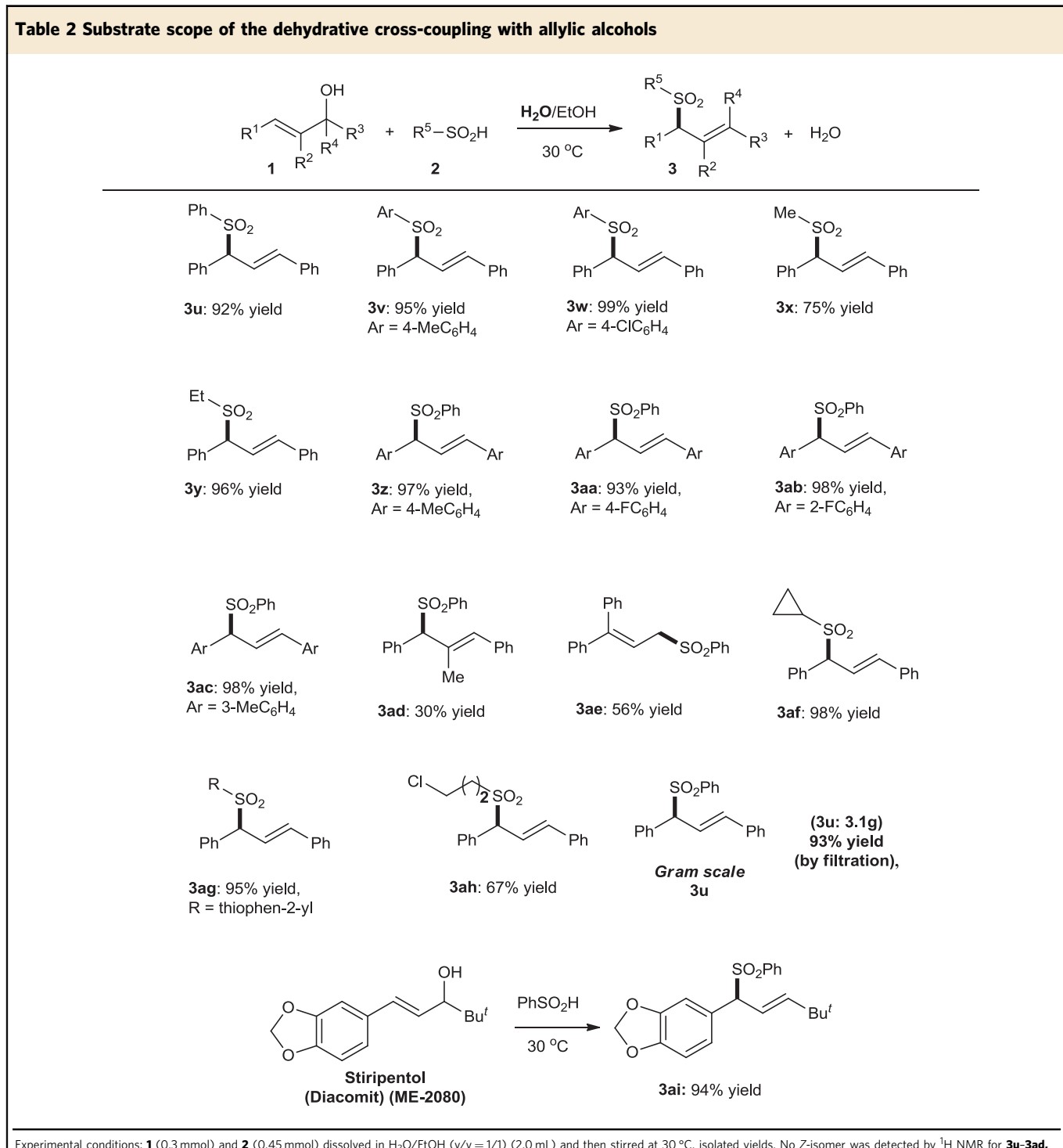

Experimental conditions: **1** (0.3 mmol) and **2** (0.45 mmol) dissolved in $H_2O$/EtOH (v/v = 1/1) (2.0 mL) and then stirred at 30 °C, isolated yields. No Z-isomer was detected by $^1$H NMR for **3u**–**3ad**, **3af**–**3ai**

can be conveniently modified by our developed methodology (**3ai**). This reaction might open new perspectives on revealing the action mechanisms of Stiripentol in human body.

**DFT calculations**. With the attempt to get more insight into the reaction mechanism, density functional theory (DFT) calculations was conducted at the B3LYP/6-31+G(d) level (Supplementary Tables 4–16, Supplementary Methods ). MBH alcohol and benzenesulfinic acids were selected as model for the mechanism

examination (Fig. 3). According to the calculation, the reaction might proceed through one-step process. The cooperative effect existed between hydroxyl group of allylic alcohol and benzenesulfinic acids via hydrogen bond. In the transition states **TS** and **TS'**, the hydrogen bond[30] not only accelerates the addition process by enhancing the nucleophilic activity of benzenesulfinic acids, but also activates the C−OH bond and results in its elimination in a water manner. Compared with **TS'**, **TS** is favored with a free energy of activity of 32.6 kcal mol$^{-1}$ with generating

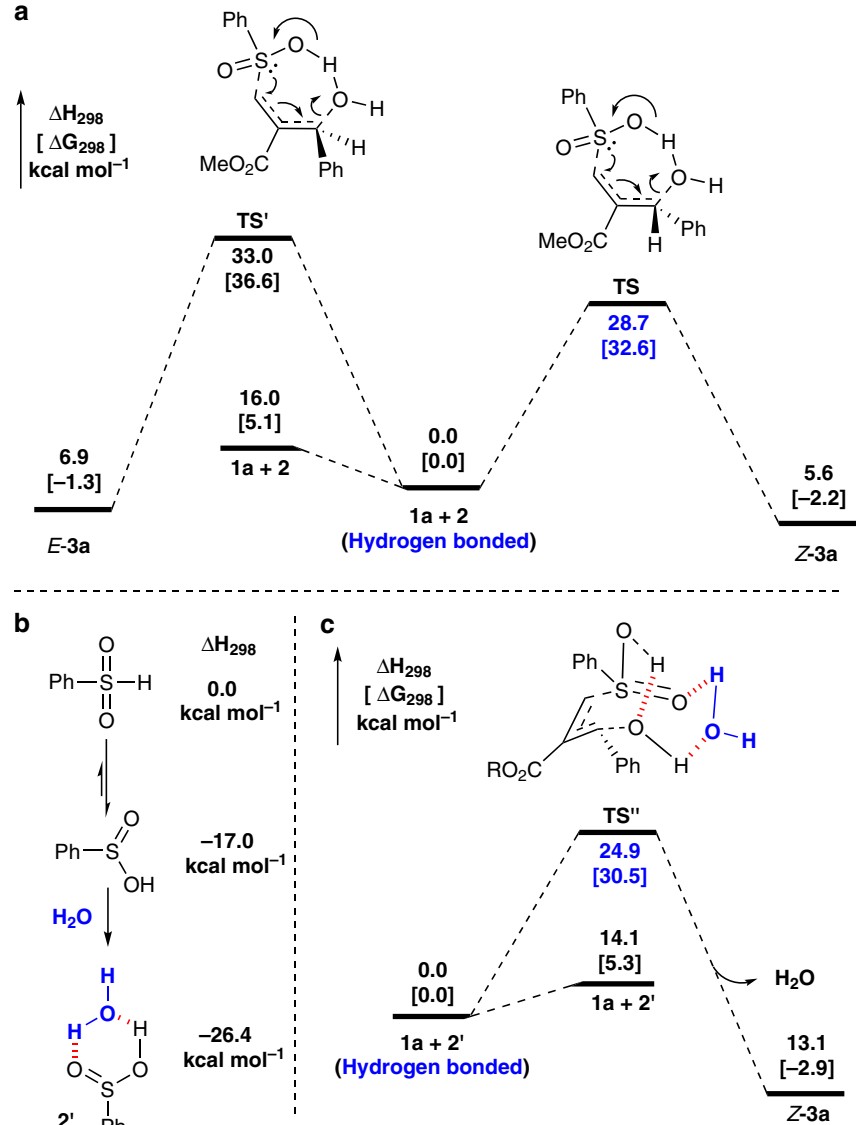

**Fig. 3** B3LYP/6-31+G(d) computed energy profile for the dehydrative cross-coupling reaction. Hydrogen bonded **1** and **2/2′** were set as references for the 0.0 energy. **a** Reaction proceed through one-step process. *Z*-isomer is favored in energy. **b** Benzenesulfinic acids can be stabilized by water forming a six-member ring format **2′**. **c** An energetically more favored bridged-ring transition states **TS″** was formed with the assistant of water

allylic sulfones with *Z*-selectivity. As is shown in Fig. 3b, water dramatically stabilized the benzenesulfinic acids by forming a six-member ring format, **2′**. Moreover, an energetically more favored bridged-ring transition state **TS″** was formed with assistance of water, which, to some extent, can account why water could dramatically enhance the overall performance of this reaction.

**Allylation of complex sulfinic acids**. The results in Tables 1 and 2 showed the excellent tolerance of the dehydrative coupling toward a variety of functional groups, including ester, nitro, alkyl, halides, cyan, and formyl. This might allow the late-stage allylation of complex sulfinic acids. With the assistance of *p*-toluenesulfonic acid, the dehydrative cross-coupling between hypotaurine and allylic alcohol afforded desired allylic sulfone **4** in quantitative yield (Fig. 4a). Importantly, cysteine sulfinic acid can also be successfully ligated using our developed method (Fig. 4b). Moreover, under acidic conditions the sulfinate can react with allylic alcohol well with the generation of

Etoricoxib (MK-0663) analog **6** in 75% yield (Fig. 5). Thus, it can safely come to the conclusion that the water-promoted C–S bond construction methodology will shed light on biological chemistry and pharmacology study such as the rational drug design concerning cysteine metabolism, beside organic synthetic community itself.

## Discussion

In summary, we have developed an unprecedented dehydrative cross-coupling strategy for the synthesis of a wide variety of allylic sulfones. This method works with a wide-spectrum allylic alcohols and sulfinic acids affording the allylic sulfones in high yield and selectivity. Both linear and branched allylic sulfones can be obtained efficiently in an environmental benign manner under physiological conditions (pH 7, aqueous media, metal-free, and room temperature). In most cases, the products precipitated out after the reactions and can be easily obtained by simple filtration without the need to perform tedious organic extraction and

**Fig. 4** Direct allylation of hypotaurine and cysteine sulfinic acid. **a** TsOH (4-methylbenzenesulfonic acid) was believed to destroy intramolecular hydrogen bonding of Hypotauring. **b** The dr value of **5** was determined by ¹H NMR

**Fig. 5** Preparation of Etoricoxib (MK-0663) analog. HCl hydrochloric acid (HCl, 37%). This reaction performed at 30 °C for 72 h

column chromatography. Water has been found to be critical for this reaction and its role in this reaction was preliminary investigated by DFT calculation which indicated that multi-hydrogen bonds were crucial for this transformation. This investigation might be helpful in revealing the action mechanisms of drugs involving allylic alcohols moieties and will shed light on cysteine metabolism related investigation and drugs design in the future.

## Methods
**Synthesis of 3a**. Benzenesulfinic acid (64 mg, 0.45 mmol) and methyl 2-((4-bromophenyl)(hydroxy)methyl)acrylate (81 mg, 0.3 mmol) were dissolved in aqueous media 2.0 mL (ethanol/deionized water, v/v = 1/1) and stirred vigorously at 30 °C for 48 h. After complete conversion, the desired allylic sulfones **3a** precipitated from the solvent, which can be obtained directly via filtration in 75% yield (88.5 mg). Alternatively, **3a** also can be obtained via column chromatography (Petroleum ether (bp: 60–90 °C)/ethyl acetate = 5:1) in 91% yield (108 mg).

**Purification method for gram-scale reaction (3u)**. Benzenesulfinic acid (2.13 g, 15 mmol) and (E)-1,3-diphenylprop-2-en-1-ol (2.1 g, 10 mmol) were suspended in aqueous media 40 mL (ethanol/deionized water, v/v = 1/1) and stirred vigorously at 30 °C for 3 h. Corresponding allylic sulfone precipitated from the solvent and then filtered and dried under vacuum to yield pure **3u** as white solid (3.10 g, 93% yield).

**Data availability**. Additional data supporting the findings described in this manuscript are available in Supplementary Methods. All of the DFT calculations were performed with the Gaussian 09 program package version A.02 at the B3LYP levels of theory with the 6-31+G(d) basis set used. For the details, see Supplementary Methods and Supplementary Tables 4–16. For full characterization data and ¹H and ¹³C NMR spectra of new compounds and experimental details, see Supplementary Methods and Supplementary Figs. 2–77. The supplementary crystallographic data for this paper could be obtained free of charge from the Cambridge Crystallographic Data Centre (**3a**: CCDC 1573430) via https://www.ccdc.cam.ac.uk/structures/Search?Ccdcid=1573430. All other data are available from the authors upon reasonable request.

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

## Acknowledgements

We greatly acknowledge financial support by the National Natural Science Foundation of China (21702108), the State Key Program of National Natural Science Foundation of China (21432009), Natural Science Foundation of Jiangsu Province, China (BK20160977), Nanjing Tech University and the SCIAM Fellowship by Jiangsu National Synergetic Innovation Center for Advanced Material.

## Author contributions

J.W., P.X., Y.L., J.F., X.W., W.F., and Z.S. performed the experiments and analysis the data. P.X. conducted the DFT calculations. P.X. and T.-P.L. design and directed the project and wrote the manuscript. All the authors discussed the results and commented on the manuscript.

## Additional information

**Competing interests:** The authors declare no competing interests.

