## [peer review file(PDF 345 kb) · Nature Communications]

Reviewer #1 (Remarks to the Author):

In this manuscript, Teck-Peng Loh et al. report an experimental and theoretical study about water-promoted C-S bond formation reactions. Although the electrophilic activation by hydrogen bond formation is not new, see the C-C bond activation by hydrogen bonds in *J. Org. Chem.* 2003, 68, 8662, the amicable reaction conditions and the good yields obtained for a wide series of substrates permit its publication in *Nature Communications* after some minor changes:

Comments:

- The allylic alcohols used as reagents are chiral molecules. The authors work with a racemic mixture or with a pure enantiomer.
- Analysis of the Z/E reaction mixtures indicates that the Z isomer is obtained in a relation > 95%. This product is obtained by a stereospecific or by a stereoselective reaction?
- In figure 4, the reference for the 0.0 energy is not the reagents, 1a+2, but a molecular complex of lower energy in which 1a and 2 are hydrogen bonded.
- The arrows given in Figure 4 are confusing because they suggest a concerted mechanism, see line 133. In reactions involving the formation and rupture of several bonds, the process can not be concerted.

Minor changes:

- line 131, (DFT) calculations.
- line 133, ... through one-step process
- line 148, B3LYP/6-31+G(d) computed energy profile...
- lines 173 and 255, DFT calculations

Reviewer #2 (Remarks to the Author):

Allylic sulfones are of great interest to the pharmaceutical industry. The synthetic methodologies to prepare this class of compounds has been mainly via transition metal catalysis under acidic conditions. Recent advances in C-S bond formation by Reddy and Hu (ref 23) were significant in the field but suffer from toxic and expensive reagents. When considering the synthesis of the ArSO₂CN as part of a green chemistry analysis, and the by-products formed in that method, a worthwhile goal is to find a greener alternative reagent.

This manuscript presents a green synthesis of allylic sulfones based on atom economy, solvent optimisation, workup procedure and energy efficiency (reaction temperature). A large number of examples are presented that showcase the wide substrate scope of this reaction. This includes multigram examples, sulfinic acid substrates based on natural products including aminoacids, and close analogues for medicinal chemistry project.

The introduction includes relevant and appropriate review and is clearly written. The work is of great interest to the organic chemistry, green chemistry and synthetic medicinal chemistry communities. The recommendation is to accept this manuscript.

Major comments

What is the purity of product which precipitates? Is this the same purity as product isolated by FC?

Have the authors investigated conditions to increase yield by precipitation of the product? i.e. antisolvent (inc additional water) or reducing solvent volume.

The method using FC states in ESI that the precipitate is dissolved in an organic solvent and washed. Considerably less solvent would be required if the precipitate was separated, and only the supernatant was extracted to isolate product. The precipitate can be dissolved in a minimum amount of solvent and washed as normal, then both organic phases combined for drying etc. This assumes the precipitated product also requires FC, else the initial separation/filtration is sufficient.

Figure 2. Reference numbers need correction.

Compound numbering requires checking and correction, for instance 5 and 6 in line 155-157 and in Figure 5 and 6.

Figure 6 shows only one example. Therefore an analogue not analogues.

Reviewer #3 (Remarks to the Author):

Loh and coworkers described in this manuscript a new and efficient protocol for the synthesis of allylic sulfones. Although there are some methods reported before for the synthesis of allylic sulfones, this method described here is unique. They simply treated allylic alcohols and sulfinic acids in a mixture of 1:1 water and ethanol, and the dehydrative cross-coupling reaction occurred at 30 °C to afford both linear and branched allylic sulfones in good to excellent yields and with high regio- and stereoselectivity. The substrate scope is broad. The reaction conditions are environmentally benign. Moreover, in many cases the products precipitated out as solid, and they could be collected by simple filtration. The manipulation is much simplified by avoiding tedious organic extraction and column chromatography. They also carried out DFT calculation to elucidate the critical role of water in this reaction. Multi-hydrogen bonds were found to be crucial for this transformation. Overall, the reaction described here is an ideal method to access allylic alcohols and should attract much attention from chemistry community. This reviewer recommends publication of this manuscript in Nature Communications.

List of Responses

Dear Editors and Reviewers:

Thank you for your letter and for the reviewers' comments concerning our manuscript entitled "Water-promoted C-S bond formation reactions" (NCOMMS-18-00490-T). Those comments are all valuable and very helpful for revising and improving our paper, as well as the important guiding significance to our researches. We have studied comments carefully and have made correction which we hope meet with approval. Revised portion are marked in blue in the paper. The main corrections in the paper and the responds to the reviewer's comments are as flowing:

Responds to the reviewer's comments:

Reviewer 1:

1. **Comment:** (The allylic alcohols used as reagents are quiral molecules. The authors work with a racemic mixture or with a pure enantiomer.)

Response: We are sorry for our negligence of mention this point. In this investigation, all the allylic alcohols were used with **racemic mixtures**. The Morita-Baylis-Hilman alcohols were preparation according to the known methods (J. Feng, X. Lu, A. Kong & X. Han. *Tetrahedron* **63**, 6035-6041 (2007)). Other allylic alcohols was also prepared according to literature (X.-D. Li, L.-J. Xie, D.-L. Kong, L. Liu, L. Cheng, *Tetrahedron* **72**, 1873-1880 (2016)).

2. **Comment:** (Analysis of the Z/E reaction mixtures indicates that the Z isomer is obtained in a relation > 95%. This product is obtained by a stereospecific or by a stereoselective reaction ?)

Response: In this investigation, when Morita-Balylis-Hillman alcohols were selected as starting material, Z-isomer was major while a trace amount of E-isomer can be detected (for **3h**, **3i**, **3o**). Thus, judging by combing these results and DFT calculations, we come to the conclusion that it is a stereoselective rather than stereospecific reaction.

3. **Comment:** (In figure 4, the reference for the 0.0 energy is not the reagents, 1a+2, but a molecular complex of lower energy in which 1a and 2 are hydrogen bonded.)

Response: Thanks for the constructive suggestions. In the revised manuscript, the structure of hydrogen bonded (**1a** and 2, **1a** and 2') were optimized by B3LYP/6-31+G(d) and set as reference for the 0.0 energy. The detail information was added to the Supplementary Tables 6, 14 and 16. The revised Figure 4 was as follow:

4. **Comment:** (The arrow given in Figure 4 are confuse because they suggest a concerted mechanism, see line 133. In reactions involving the formation and rupture of several bonds, the process can not be concerted.)

Response: We sincerely accept the comments. In the revised manuscript, in line 133 “---might proceed through concerted process.” was replaced by “might proceed through one-step process”.

5. Reviewer mentioned other **Minor changes:** “line 131, (DFT) calculations”; “line 133, ... through one-step process; line 148, B3LYP/6-31+G(d) computed energy profile...”; “lines 173 and 255, DFT calculations”

Response: Thank you for your positive suggestions.

line 131, 173, 255, the words “calculation” were corrected as “calculations”.

line 133, “... through concerted process” was corrected as “... through one-step process”.

The title of Figure 4 was corrected as “B3LYP/6-31+G(d) computed energy profile for the dehydrative cross-coupling reaction”

The literature “Domingo, L. R. & Andres, J. Enhancing reactivity of carbonyl compounds via hydrogen-bond formation. A DFT study of the Hetero-Diels–Alder reaction between butadiene derivative and acetone in chloroform. J. Org. Chem. 68, 8662-8668 (2003).” was cited as Ref. 30.

Special thanks to you for your good comments.

Reviewer 2:

1. **Comment:** (What is the purity of product which precipitates? Is this the same purity as product isolated by FC?)

Response: Thank you very much for the kind reminding. **3f** and **3u** was generated from MBH alcohols and 1,3-diarylprop-2-en-1-ols respectively. In addition, both of them were obtained in a lower yield than their analogues. Moreover, the reaction for generating **3f** and **3u** was also conducted in the gram scale. Thus, it will be representative samples for check the purity of precipitates. The reaction between (E)-1,3-diphenylprop-2-en-1-ol and benzenesulfonic acid to give **3u**, which precipitated as white solid and can be obtained directly by filtration and dried under vacuum. As checked by NMR, the precipitate was **3u** and with same purity as the product isolated by FC. On the other hand, allylic sulfones **3f** precipitated as sticky oil at reaction temperature (probably due to the lower melting point 64-66 °C, the purity of the sticky oil is over 95% by NMR), which was followed by recrystallization from 10 mL EtOH/H₂O (V/V = 1/1) afforded a white solid **3f** (3.01g). These detail information was also added to the revised Supplementary Information.

2. **Comment:** (Have the authors investigated conditions to increase yield by precipitation of the product? i.e. antisolvent (inc additional water) or reducing solvent volume.)

Response: We have investigated conditions to increase yield by precipitation of the product. When the aqueous media (0.5 mL EtOH with 1.5 mL H₂O) was used, the poor solubility of starting materials disfavored the transformation and delivered **3a** in lower (72%) yield. Thus, additional water can't give positive result.

On the other hand, the reaction proceeded in higher concentration (EtOH (0.5 mL)/H₂O(0.5 mL); EtOH (0.25 mL)/H₂O(0.25 mL)) was then investigated. As is

shown in the following scheme, reaction performed well while **3f** precipitated as sticky oil in a higher yield, albeit the purity of sticky oil is slight lower (trace amount of starting materials can be detected by NMR) than under identity conditions (comparable purity as product isolated by with FC). For **3u**, the concentrations have little effect on the yield and purity of precipitates. Thanks again for review's suggestions.

3. **Comment:** (The method using FC states in ESI that the precipitate is dissolved in an organic solvent and washed. Considerably less solvent would be required if the precipitate was separated, and only the supernatant was extracted to isolate product. The precipitate can be dissolved in a minimum amount of solvent and washed as normal, then both organic phases combined for drying etc. This assumes the precipitated product also requires FC, else the initial separation/filtration is sufficient.)

Response: Thanks for your constructive suggestions. In the purified process, the aqueous media was extracted with ethyl acetate (2*5 mL) rather than (2*25 mL). The correction was made in the revised supplementary methods. According to your suggestion, we conducted the model reaction (for **3a** and **3u**) and purified with less organic solvent. During the purification for **3a**, the precipitate was firstly separated, and the supernatant was extracted with ethyl acetate (2*2 mL). Then the precipitate then dissolved in 1 mL ethyl acetate and combined with extract solution before concentrated by rotary evaporation. The crude product was then purified *via* column chromatography to give the corresponding products **3a** in 88% yield. For the **3u**, the initial separation/filtration is sufficient to get purity **3u** in 87% yield.

Thanks again for the environmental concerning suggestion and will help us a lot in our future investigation.

4. **Comment:** (Figure 2. Reference numbers need correction. Compound numbering requires checking and correction, for instance 5 and 6 in line 155-157 and in Figure 5

and 6. Figure 6 shows only one example. Therefore an analogue not analogues.)

Response: Thank you for the careful corrections. We carefully rechecked the references and compounds numbers in the revised manuscript.

In figure 2, “ref.37-41” was replaced by “ref.(17-20)”, “ref. 44” was replaced by “ref. 23”.

In lines 154-155, “--- cysteine sulfinic acid can also be successfully ligated using our developed method.” was corrected as “--- cysteine sulfinic acid can also be successfully ligated using our developed method (5).”.

In lines 156-157, “---generation of Etoricoxib (MK-0663) analogues 5 in 75% yield (Fig. 6).” was corrected as “---c generation of Etoricoxib (MK-0663) analogue 6 in 75% yield (Fig. 6).”.

In Figure 6, “analogue” was replaced by analogue both in title and molecular.

In lines 72, ref. “(5)” was corrected as Ref. “(4)”

Reviewer 3:

Thank you for the affirmations of this investigation.

We tried our best to improve the manuscript and made some changes to comply with Nature Communications format requirements. These changes will not influence the content and framework of the paper. And here we did not list the changes but marked in blue in revised paper.

We appreciate for Editors and Reviewers’ warm work earnestly, and hope that the correction will meet with approval.

Once again, thank you very much for your comments and suggestions.

Yours sincerely,

Teck-Peng Loh

Corresponding author:

Name: Teck-Peng Loh

E-mail: teckpeng@ntu.edu.sg

reviewer 1

In this revised version, Xie et al. have considered the different questions that I raised. Consequently, in my opinion, the manuscript can be published in Nature Communications in the present form.

reviewer 2

The authors have addressed all points raised by reviewers. Recommendation is to accept.